# Isolation, identification and antibiotic resistance profile of thermophilic *Campylobacter* species from Bovine, Knives and personnel at Jimma Town Abattoir, Ethiopia

**Motuma Debelo**[1]*, **Nezif Mohammed**[1], **Abebaw Tiruneh**[2], **Tadele Tolosa**[1]

**1** School of Veterinary Medicine, College of Agriculture and Veterinary Medicine, Jimma University, Jimma, Ethiopia, **2** School of Medical Laboratory Sciences, Institute of Health, Jimma University, Jimma, Ethiopia

* motuma2017ju@gmail.com

**Data Availability Statement:** All relevant data are available in the paper and Supporting information files.

## Abstract

Thermophilic *Campylobacter* species are common cause of animal and human bacterial diseases with growing resistance to antimicrobials. The aim of this study was to determine the prevalence and antimicrobial susceptibility pattern of *Campylobacter* species from bovine, knives and personnel in Jimma Town, Ethiopia. Faecal samples and carcasses swabs were collected from cattle systematically selected from the annual plan of Jimma Municipal Abattoir. Personnel hand and knife swabs were collected after slaughtering each selected cattle. A cross-sectional study with systematic sampling method was conducted from October 2019 to September 2020 for the isolation, identification and antimicrobial susceptibility pattern of thermophilic *Campylobacter* species. Isolation and identification of *Campylobacter* species were performed according to the techniques recommended by the International Organization for Standardization, and *in vitro* antibiotic susceptibility testing was screened using the standard agar disc diffusion method as recommended by Clinical and Laboratory Standards Institutions. A total of 684 samples (171 samples from faeces, carcasses, knives and personnel hands, were collected independently). The overall prevalence of thermophilic *Campylobacter* species was 5.6% (38/684). Majority of the isolates were from faecal samples (12.9%, n = 22) followed by carcass swabs(4.1% n = 7), knife swabs(3.5% n = 6) and personnel hand swabs(1.8% n = 3). Isolated and identified species of *C.jejuni*, *C. coli* and *C. lari* accounted for 63.2%, 23.7% and 13.2%, respectively. The isolated *Campylobacter* species were found to be resistant to Cephalothin (100%), Ampicillin (60.5%), Cefotaxime (60.5%), Chloramphenicol (47.4%) and Tetracycline (42.1%). On the other hand, the isolates were susceptible to Nalidixic acid (86.8%), Ciprofloxacin (86.8%), Sulphamethazole (84.2%), Ceftriaxone (78.9%), Clindamycin (68.4%) and Cefixime (65.8%). 84.2% of the isolates showed multi-drug resistance for three-to-six drug classes. All the *C. lari* isolates were multidrug resistant. All the three isolated species of *Campylobacter* were resistant to Cephalothin, and most were multidrug resistant. Isolation of *Campylobacter* species from faecal, carcass, knife and hand swabs revealed possible risk of contamination and exposure to

**Funding:** Authors who received each award; Motuma Debelo, Nezif Mohammed, Abebaw Tiruneh and Tadele Tolosa No Specific Grant numbers awarded to each author. This research work was funded by Jimma University College of Agriculture and Veterinary Medicine. The funders had no role in study design, data collection and analysis, decision to publish, or preparation of the manuscript.

**Competing interests:** The authors have declared that no competing interests exist.

**Abbreviations:** AMP, Ampicillin; AMR, Antimicrobial resistance; C, Chloramphenicol; CFM, Cefixime; CH, Cephalothin; CIP, Ciprofloxacin; CO2, carbon dioxide; CRO, Ceftriaxone; CTX, Cefotaxime; DA, Clindamycin; H2S, hydrogen sulphide; mCCDA, modified Cefoperazone Charcoal Deoxycholate Agar; MDR, multi-drug resistant; NA, Nalidixic acid; OR, odd ratio; SXT, Sulfamethoxazole trimethoprim; TE, Tetracycline; VDFACA, Administration and Control Authority of Ethiopia.

Campylobacter infection of those who consume raw meat. Therefore, enactment of hygienic practices during the slaughtering process, proper handling and cooking of meat and awareness creation on jurisdictional antibiotic usage are required to avoid *Campylobacter* infection.

## Introduction

*Campylobacters* are gram negative, non-spore forming, slender, spiral to curved, rod-shaped bacteria, and infect commonly warm blooded animals. *Campylobacter* species are among the major causes of food-borne bacterial diseases of significant public health and veterinary importance. *Campylobacter* species are known to cause gastroenteritis in human, caused predominantly by *C. jejuni*, followed by *C. coli*, *C. lari* and *C. fetus*. It remained as global health concern as the group *Campylobacter* has zoonotic potential, large range of reservoir hosts, and environmental persistence [1, 2].*Campylobacter* is the fourth (next to rotavirus, typhoid fever and cryptosporidiosis) leading cause of diarrheal diseases of human with estimated 7.5 million global disability adjusted life years. Its transmissions to humans is mainly through food-borne, from food animals and animal products [3]. Moreover, humans may acquire the infection from cross-contamination of ready-to-eat meat by knives, cutting boards or contaminated hands and/or improper cooking. Individuals working in animal facilities and animal food factories may also acquire the infection via occupational exposure [3–5].

Thermophilic *Campylobacter* species are commonly isolated from poultry and livestock species such as cattle, swine, and sheep, but poultry is recognized as the primary source of infection for people [6, 7]. The role of cattle in the epidemiology of *Campylobacter* is not known fully and likely a complex web of transmission exists between people, poultry, cattle, other livestock, wild reservoir hosts, and the environment. However, cattle are identified as a source of infection for people and water is implicated as the spread of *Campylobacter*, in the environmental in particular. [1, 8, 9].

On the other hand, antimicrobial resistance (AMR) is recognized as a One Health challenge because of the rapid emergence and dissemination of resistant microbial genes among humans, animals and the environment. The magnitude of AMR is spreading more rapidly in the developing countries due to uncontrolled use of different antibiotics [10, 11]. *Campylobacters* also harbor AMR genes with the capability of horizontal transfer between pathogenic and commensal microorganisms, which could lead possibly to the emergence of multi-drug resistant (MDR) microorganisms [12]. Hence, *Campylobacter* species are considered as a member of global priority pathogens due to their patterns of AMR. The susceptibility of *Campylobacter* to available drugs is tending decreased and continuous assessment of AMR at different sites and regions is essential [13, 14].

Studies on thermophilic *Campylobacter* species from human and domestic animals illustrated zoonotic importance of the disease in Ethiopia [15–18]. However, limited studies reported the occurrence and susceptibility testing of *Campylobacter* strains to antimicrobials on humans, food animals and foods of animal origin [16–18]. Moreover, updated information is needed on the sources of infection, disease prevalence, and antibiotic susceptibility status in different localities. Therefore, the aim of this study was (i) to isolate and identify thermophilic *Campylobacter* species from live bovine faeces, its carcass, knife swabs and personnel hand swabs, (ii) to determine antibiotic susceptibility of the isolates at Jimma Town Municipal Abattoir. Our research finding determines the current status of *Campylobacter* species in Jimma

Town Abattoir, source of contamination and level of antibiotic resistance. It also serves as a baseline research to study source of contamination, to consider the level of abattoir hygiene and to initiate regular inspection of abattoirs for *Campylobacter* infection and antimicrobial resistance patterns.

## Materials and methods

### Study area

The study was conducted at Jimma Municipal Abattoir in Jimma Town. Jimma Town is located in Southwest of Ethiopia, 353 km away from the capital city, Addis Ababa. Jimma Town is surrounded by rural districts that mainly rearing cattle, sheep, goat and poultry. The town is geographically located at 7˚41′N latitude, 36˚50′E longitude, the average altitude of 1,780 meters above sea level, and commonly characterized as temperate weather with mean annual maximum and minimum temperature of 30˚C and 14˚C, respectively. The annual rainfall ranges between 1138mm and 1690mm [19]. Jimma Town has one municipality abattoir and about 140 meat retailers which receive directly slaughter service from the abattoir. The abattoir gives service for 207,573 people (projected population in 2021) in the town with 50.52km$^2$ area. Jimma Town Municipal Abattoir gives slaughter services of cattle, sheep and goats. The origin of animal for slaughter is mainly from the surrounding districts of Jimma Zone and other nearby areas.

### Study design and period

Faecal samples and carcasses swabs were collected from cattle systematically selected from the annual plan of Jimma Municipal Abattoir in 2019/2020. Personnel hand and knife swabs were collected after slaughtering each selected cattle. Cross sectional study design with systematic sampling method was used. The study was conducted for a year (October 2019 to September 2020) at Jimma Town Municipal Abattoir.

### Sample size and sampling

The sample size was calculated using single population proportion formula [20]. Previous bovine thermophilic *campylobacter* prevalence in Jimma Zone is 12.7% [13], 5% of desired absolute precision and 95% confidence interval.

$$n = 1.96^2 \text{Pexp} \, (1 - \text{Pexp})/d^2 = 1.96^2(0.127)(1 - 0.127)/(0.05)^2 = 0.4259/0.0025 = 171$$

Where, n = required sample size, Pexp = expected prevalence, d = desired absolute precision.

The estimated annual plan of cattle slaughtering in 2019/2020 at Jimma Town Municipal Abattoir was 2,500 cattle and every 15$^{th}$ (K) cattle was selected consecutively as the study subject. Totally, 684 samples were taken from apparently healthy bovine (faecal), their carcasses swab, personnel hand swab and knife swab, 171 samples for each sample type. Swabs from the personnel hands and knives were taken from each selected cattle after slaughtering.

### Sample collection and transportation

From each selected bovine, about two gram faecal sample was obtained rectally from untouched glove center using swabber stick and s carcass swabs were taken from surface meat of flank and briskets areas (pooled swabs from both sites) using cotton tipped swabs. Samples from the environment, personnel hands and knives were taken aseptically with sterile cotton tipped swabs. All fecal samples were placed into a sterile screw capped container containing 9 ml Cary-Blair transporting media. In case of swab samples, each cotton wool swab was

moistened and rubbed on sites continuously for 30 seconds and transferred to a sterile screw-capped test tube containing 10 ml of Cary-Blair transporting media. Then, the samples were transported to Jimma University College of Agriculture and Veterinary Medicine (JUCAVM) Laboratory using ice box at 2–8˚C. All swabs were processed within 3 hours of collection.

## Isolation and identification of thermophilic *campylobacter species*

Isolation and identification of thermophilic *campylobacter* species were performed according to the techniques recommended by the International Organization for Standardization [21]. Faecal samples, carcass swabs, knife swabs and personnel hand swabs from transport medium was streaked on to *Campylobacter* blood free selective agar base called modified Cefoperazone Charcoal Deoxycholate Agar (mCCDA from Oxoid Ltd.) that supplemented with SR155H supplement, and non-blood free *Campylobacter* selective agar base with campylobacter supplement (Blaserwanger: from HIMEDIA). Streaked plates were incubated at 42˚C in anaerobic jar under a micro-aerophilic atmosphere condition (84% N2, 10% $CO_2$, 6% $O_2$) produced from gas generating sachets (Campy-Gen TM; Oxoid Ltd.) for 48 hours and plates with no growth were incubated in a micro-aerophilic condition for additional 24 hours (totally for 72 hrs.).

After two to three days incubation under micro-aerobic condition, mCCDA plates were assessed visually for presence of *Campylobacter* species colonies. One presumptive *Campylobacter* colony from each selective agar plate was sub-cultured and tested by standard microbiological and biochemical procedures. Preliminary identification of thermophilic *Campylobacter* species was performed based on microscopy to see characteristic darting motility by preparing wet smear or saline wet mount, with the iris diaphragm closed effectively to contrast the field. Gram stain was performed to identify the characteristics of *Campylobacter*, Gram negative with an 'S' shaped and gull wing appearance. Moreover, Oxidase test and Catalase test were performed for the identification of thermo-tolerant *Campylobacter* genera.

The colonies of thermophilic *Campylobacter* from blood agar medium were picked up with a sterilized cotton swab and put into small tubes containing storage medium (Brain- heart-infusion broth medium) for identification. Triple sugar iron agar ($H_2S$ production), Hippurate hydrolysis test, resistance to Cephalothin (30 μg disc) and susceptibility to Nalidixic acid (30 μg) disk were evaluated and interpreted. The parameters formed were based on the basis for the identification of *Campylobacter* species [21, 22].

## Antibiotic susceptibility pattern of *campylobacter* species

The isolated thermophilic *Campylobacter* species were screened for *in vitro* antimicrobial susceptibility using the standard agar disc diffusion method as recommended by Clinical and Laboratory Standards Institutions [23] for eleven antimicrobial agents on Mueller-Hinton agar (Oxoid Ltd.) supplemented with 5% sheep blood. Antimicrobial disks were selected based on availability and utilization of the drugs in Ethiopia for animal treatment. The information was collected from Veterinary Drug and Animal Feed Administration and Control Authority of Ethiopia (VDFACA). The 11 different antibiotic discs, with their concentrations given in parenthesis, used in the antibiogram testing were; Ampicillin (AMP) (10μg), Nalidixic acid (NA) (30μg), Tetracycline (TE) (30μg), Ceftriaxone (CRO) (30μg), Cefixime (CFM) (5μg), Cefotaxime (CTX) (30μg), Clindamycin (DA) (10μg), Sulfamethoxazole trimethoprim (SXT) (25μg), Ciprofloxacin (CIP) (10μg), Chloramphenicol (C) (30μg), and Cephalothin (CH) (30μg).

Three to four morphologically identical colonies of bacteria were picked and suspended in sterile normal saline. Turbidity of the broth culture was measured with turbido-meter with in

absorbance reading range of 0.08 to 0.1 at 625nm (equivalent to 0.5 McFarland turbidity standards). A loop full of the bacterial suspension was placed at the center of Muller Hinton agar media (Oxoid, Ltd.) supplemented with 5% sheep blood and evenly spread using sterile cotton tipped applicator. After drying, antimicrobial-impregnated disks were appropriately placed on the surface of the agar using sterile forceps by carefully removing one disk from the cartridge. Eight (8) disks were placed on a 150-mm plate (petri dishes) not closer than 24 mm (center to center) and the rest three on small 60-mm petri dishes on the Muller Hinton agar plate and incubated at 42˚C for 48 hours in anaerobic jar using microaerophilic condition generating sachet (CampyGen™ 2.5L Oxoid Ltd.).

For each thermophilic *Campylobacter* isolate, the zone of inhibition around each of the 11 antibiotic disks was measured to the closest millimeter. Interpretive criteria [24] for *Campylobacter* susceptibility testing recommended by the Clinical and Laboratory Standards Institute and zone diameter breakpoints for Enterobacteriaceae were used [25]. Each isolate was classified as susceptible, intermediate, or resistant to each of the 11 antibiotics using the published zone diameter standards for *Campylobacter* [24–26]. Ingredient activity of the disk was checked with standardized reference strain of *E. coli* (ATCC 25922); sensitive to all the antimicrobial drugs tested by Ethiopian Public Health Institute (EPHI) was used as a control. For each, antimicrobial inhibition zone was measured by using digital Vernier calipers and inhibition zone of each antimicrobial was classified (resistant, intermediate, or susceptible) according to Clinical and Laboratory Standards Institutions [23].

### Ethics statement

Ethical approval was obtained from Jimma University College of Agriculture and Veterinary Medicine Minutes of Animal Research Ethics and Review committee. Permission was sought from Jimma Town Municipal Abattoir and informed consent was obtained from individuals providing slaughtering service to take personnel hand and knife swabs.

### Data management and analysis

Laboratory data were entered to Microsoft Excel spread sheet, cleaned and transported to STATA_MP version 12. Descriptive statistics was used for calculating percentage and frequency distributions, and odds ratio (OR) was used to evaluate the association of variables at 95%CI and $p < 0.05$ was considered as statistically significant.

## Results

### Prevalence of thermophilic *Campylobacter* species

From cultured 171 bovine faecal samples and 171 bovine carcass swabs, 22 (12.9%) and 7 (4.1%) were positive for thermophilic *Campylobacter* species, respectively. Similarly, from 171 knife swab and 171 personnel hand swab cultures, 6(3.5%) and 3(1.8%) had thermophilic *Campylobacter* species growth, respectively. The overall prevalence of thermophilic *Campylobacter* was 5.6% (38/684) (Table 1). Prevalence of *Campylobacter* species detected in faecal samples was significantly higher as compared to hand swab samples with OR = 8.3(95%CI 2.4–28.2) and p = 0.001.

### Identification of thermophilic *campylobacter* species

From the total 38 isolates of faecal samples, carcass, knife and personnel hand swabs, 24 (63.2%), 9 (23.7%) and 5 (13.1%) were *C. jejuni*, *C. coli* and *C. lari*, respectively (Table 2).

**Table 1. Prevalence of thermophilic *Campylobacter* species from different sample types in Jimma Town.**

| Type of samples | Samples examined | Number of positive (%) | OR(95%CI) | P- value |
|---|---|---|---|---|
| Faecal samples | 171 | 22 (12.9) | 8.3(2.4–28.2) | 0.001 |
| Carcass swab samples | 171 | 7 (4.1) | 2.4(0.6–9.4) | 0.212 |
| knife swab samples | 171 | 6 (3.5) | 2.0(0.5–8.3) | 0.320 |
| Hand swab samples | 171 | 3 (1.8) | 1 | |
| Over all | 684 | 38(5.6) | | |

## Antimicrobial susceptibility pattern

A total of 38 thermophilic *Campylobacter* species isolated from live bovine, carcass swab and separated environmental swabs (personnel hand and knife swabs) were tested for eleven antimicrobial agents. Higher resistance was observed on Cephalothin (CH-30): 100%, Ampicillin (AMP-10): 60.5% and Cefotaxime (CTX-30): 60.5%. However, the isolates showed higher susceptibility to Nalidixic acid (NA-30): 86.8%, Ciprofloxacin (CIP-10): 86.8% and Sulphamethazole (SXT-25): 84.2% (Table 3).

Table 4 Summarizes *in vitro* antimicrobial susceptibility patterns of each *Campylobacter* species from different samples in which *Campylobacter lari* showed higher resistance for all tested drugs as compared to *C. jejuni* and *C. coli*.

## Multi-drug resistance pattern of thermophilic *campylobacter* species isolates

The 11 antimicrobial agents were from nine antimicrobial category, accordingly: Ampicillin (AMP-10): Penicillins class; /Cefixime (CFM-5), Cefotaxime (CTX-30) and Ceftriaxone (CRO-30)/: Extended-spectrum cephalosporins 3rd and 4th generation cephalosporins class; Cephalothin (CH-30): Non-extended spectrum cephalosporins 1st and 2nd generation cephalosporins class; Chloramphenicol (C-30): Phenicols class; Ciprofloxacin (CIP-10): Fluoroquinolones class; Sulphamethazole Trimethoprim (SXT-25): Folate pathway inhibitors class; Tetracycline (TE-30): Tetracyclines class; Clindamycin (DA-10): Lincosamides class; Nalidixic Acid (NA-30): Quinolone Antibiotics class. Drug resistance was recorded among isolated thermophilic *Campylobacter* species for at least one antimicrobial agent in two to seven antimicrobial classes. Multi-drug resistance (*Campylobacter* isolate showing resistance against at least to one antimicrobial agent in three or more antimicrobial classes/category) [27] was observed in all isolated species and *C. lari* showed higher MDR pattern as compared to *C. jejuni* and *C. coli*. The resistance pattern of thermophilic *Campylobacter* species isolates is summarized in Table 5.

**Table 2. Identification of the three *Campylobacter* species in different samples in Jimma Town.**

| Species of campylobacter | Sample types proportion | | | | Total |
|---|---|---|---|---|---|
| | Faecal sample | Carcass swab | Knife swab | Hand swab | |
| *C.jejuni* | 15 (68.2%) | 4 (57.1%) | 3 (50%) | 2 (66.6%) | 24 (63.2%) |
| *C.coli* | 4 (18.2%) | 2 (28.6%) | 2 (33.3%) | 1 (33.3%) | 9 (23.7%) |
| *C.lari* | 3 (13.6%) | 1 (14.2%) | 1 (16.6%) | - | 5 (13.1%) |
| *Total* | 22 (57.9%) | 7 (18.4%) | 6 (15.8%) | 3 (7.9%) | 38 (100%) |

**Table 3. *In vitro* antimicrobial sensitivity patterns of *Campylobacter* species isolates, Jimma Town.**

| Types of antibiotics | Interpretations | | |
|---|---|---|---|
| | Susceptible (%) | Intermediate (%) | Resistance (%) |
| AMP-10[1] | 9(23.7%) | 6(15.8%) | 23(60.5%) |
| CFM-5[2] | 25(65.8%) | - | 13(34.2%) |
| CTX-30[3] | 15(39.5%) | - | 23(60.5%) |
| CRO-30[4] | 30(78.9%) | 5(13.2%) | 3(7.9%) |
| CH-30[5] | - | - | 38(100%) |
| C-30[6] | 13(34.2%) | 7(18.4%) | 18(47.4%) |
| CIP-10[7] | 33(86.8%) | 2(5.3%) | 3(7.9%) |
| SXT-25[8] | 32(84.2%) | - | 6(15.8%) |
| TE-30[9] | 15(39.5%) | 7(18.4%) | 16(42.1%) |
| DA-10[10] | 26(68.4%) | - | 12(31.6%) |
| NA-30[11] | 33(86.8%) | - | 5(13.2%) |

[1]Ampicillin (AMP-10),

[2]Cefixime (CFM-5),

[3]Cefotaxime (CTX-30),

[4]Ceftriaxone (CRO-30),

[5]Cephalothin (CH-30),

[6]Chloramphenicol (C-30),

[7]Ciprofloxacin (CIP-10),

[8]Sulphamethazole Trimethoprim (SXT-25),

[9]Tetracycline (TE-30),

[10]Clindamycin (DA-10),

[11]Nalidixic Acid (NA-30)

## Discussion

Campylobacteriosis is bacterial, foodborne and zoonotic gastroenteritis disease transmitted to human mainly via undercooked or contaminated meat and animal products. Although poultry is considered as the main source of infection for human, cattle also play substantial role for transmission of the disease in Ethiopia, particularly where consumption of raw beef is a culture [28]. According to Jimma Municipal Abattoir report, majority of the animals slaughtered per day for public sale markets are cattle. In the abattoir, cattle usually inspected before slaughtering and the slaughtered cattle were considered as apparently healthy. However, campylobacteriosis is often asymptomatic and self-limiting, and missed during physical diagnosis. Indeed, *C. jejuni* and *C. fetus* are responsible to cause infertility and abortion in cattle [29, 30]. In the current study, the prevalence of thermophilic *Campylobacter* species isolates was (12.9%) in fecal samples followed by (4.1%) in carcasses swabs, (3.5%) in knives swabs and (1.8%) in personnel hand swabs. Increased prevalence of *Campylobacter* species from fecal sample is because the bacteria colonize mainly small intestine of cattle. Carcasses and other environmental samples acquired the bacterium from contamination mainly during slaughtering process and distribution to butcher shops [31]. Previous study conducted in Jimma Zone urban and rural farm animals also documented comparable prevalence (12.7%) of *Campylobacter* species from cattle fecal samples [15]. However, higher prevalence of *Campylobacter* species isolates were reported in studies conducted in Gambella and Gondar, Ethiopia [16, 18]. The inconsistency of the prevalence may be ascribed to geographical and environmental variability in the country.

**Table 4. Over all in vitro antimicrobial susceptibility pattern of *C. jejun*, *C. coli* and *C. lari* isolates from Jimma Town.**

| Drugs | Sensitivity | C. jejuni | | | | C. coli | | | | C. lari | | | |
|---|---|---|---|---|---|---|---|---|---|---|---|---|---|
| | | Fecal | Carcass | Knife | Hand | Fecal | Carcass | Knife | Hand | Fecal | Carcass | Knife | Hand |
| | | No/% | No/% | No/% | No/% | No/% | No/% | No/% | No/% | No/% | No/% | No/% | No/% |
| AMP-10[1] | S[a] | 5(33.3) | 2(50.0) | 1(33.3) | - | 1(25.0) | 0(0) | - | - | - | - | - | - |
| | I[b] | 3(20.0) | - | - | 1(50.0) | - | 1(50.0) | 1(50.0) | - | - | - | - | - |
| | R[c] | 7(46.7) | 2(50.0) | 2(66.7) | 1(50.0) | 3(75.0) | 1(50.0) | 1(50.0) | 1(100) | 3(100) | 1(100) | 1(100) | - |
| CFM-5[2] | S | 11(73.3) | 4(100) | 2(66.7) | 1(50.0) | 2(50.0) | 1(50.0) | 1(50.0) | 1(100) | 1(33.3) | - | 1(100) | - |
| | I | - | - | - | - | - | - | - | - | - | - | - | - |
| | R | 4(26.7) | - | 1(33.3) | 1(50.0) | 2(50.0) | 1(50.0) | 1(50.0) | - | 2(66.7) | 1(100) | - | - |
| CTX-30[3] | S | 5(33.3) | 1(25.0) | 2(66.7) | 2(100) | 2(50.0) | 1(50.0) | - | 1(100) | 1(33.3) | - | - | - |
| | I | - | - | - | - | - | - | - | - | - | - | - | - |
| | R | 10(66.7) | 3(75.0) | 1(33.3) | - | 2(50.0) | 1(50.0) | 2(100) | - | 2(66.7) | 1(100) | 1(100) | - |
| CRO-30[4] | S | 14(93.3) | 3(75.0) | 2(66.7) | 2(100) | 2(50.0) | 1(50.0) | 2(100) | 1(100) | 2(66.7) | - | 1(100) | - |
| | I | 1(6.7) | - | 1(33.3) | - | 1(25.0) | 1(50.0) | - | - | - | 1(100) | - | - |
| | R | - | 1(25.0) | - | - | 1(25.0) | - | - | - | 1(33.3) | - | - | - |
| CH-30[5] | S | - | - | - | - | - | - | - | - | - | - | - | - |
| | I | - | - | - | - | - | - | - | - | - | - | - | - |
| | R | 15(100) | 4(100) | 3(100) | 2(100) | 4(100) | 2(100) | 2(100) | 1(100) | 3(100) | 1(100) | 1(100) | - |
| C-30[6] | S | 6(40.0) | 1(25.0) | 1(33.3) | 1(50.0) | 1(25.0) | - | 1(50.0) | - | 1(33.3) | 1(100) | - | - |
| | I | 3(20.0) | - | 1(33.3) | - | 1(25.0) | - | 1(50.0) | - | - | - | 1(100) | - |
| | R | 6(40.0) | 3(75.0) | 1(33.3) | 1(50.0) | 2(50.0) | 2(100) | - | 1(100) | 2(66.7) | - | - | - |
| CIP-10[7] | S | 12(80) | 4(100) | 3(100) | 2(100) | 3(75.0) | 2(100) | 2(100) | 1(100) | 2(66.7) | 1(100) | 1(100) | - |
| | I | - | - | - | - | 1(25.0) | - | - | - | 1(33.3) | - | - | - |
| | R | 3(20.0) | - | - | - | - | - | - | - | - | - | - | - |
| SXT-25[8] | S | 12(80.0) | 4(100) | 2(66.7) | 1(50.0) | 4(100) | 2(100) | 2(100) | 1(100) | 2(66.7) | 1(100) | 1(100) | - |
| | I | - | - | - | - | - | - | - | - | - | - | - | - |
| | R | 3(20.0) | - | 1(33.3) | 1(50.0) | - | - | - | - | 1(33.3) | - | - | - |
| TE-30[9] | S | 7(46.7) | 1(25.0) | 2(66.7) | - | 2(50.0) | 2(100) | - | - | 1(33.3) | - | - | - |
| | I | 3(20.0) | 1(25.0) | - | 1(50.0) | 1(25.0) | - | - | 1(100) | - | - | - | - |
| | R | 5(33.3) | 2(50.0) | 1(33.3) | 1(50.0) | 1(25.0) | - | 2(100) | - | 2(66.7) | 1(100) | 1(100) | - |
| DA-10[10] | S | 7(46.7) | 4(100) | 1(33.3) | 2(100) | 4(100) | 1(50.0) | 2(100) | 1(100) | 3(66.7) | - | 1(100) | - |
| | I | - | - | - | - | - | - | - | - | - | - | - | - |
| | R | 8(53.3) | - | 2(66.7) | - | - | 1(50.0) | - | - | - | 1(100) | - | - |
| NA-30[11] | S | 15(100) | 4(100) | 3(100) | 2(100) | 4(100) | 2(100) | 2(100) | 1(100) | - | - | - | - |
| | I | - | - | - | - | - | - | - | - | - | - | - | - |
| | R | - | - | - | - | - | - | - | - | 3(100) | 1(100) | 1(100) | - |

[1]Ampicillin (AMP-10),

[2]Cefixime (CFM-5),

[3]Cefotaxime (CTX-30),

[4]Ceftriaxone (CRO-30),

[5]Cephalothin (CH-30),

[6]Chloramphenicol (C-30),

[7]Ciprofloxacin (CIP 10),

[8]Sulphamethazole Trimethoprim (SXT 25),

[9]Tetracycline (TE 30),

[10]Clindamycin (DA 10),

[11]Nalidixic Acid (NA 30),

[a]S = Sensitive,

[b]I = Intermediate,

[c]R = Resistant

**Table 5. Multidrug-resistance patterns of *Campylobacter* species isolated from Jimma Town municipal abattoir.**

| Resistance category | Resistance Pattern | *C. jejuni* (n = 24) | | *C. coli* (n = 9) | | *C. lari* (n = 5) | |
|---|---|---|---|---|---|---|---|
| | | No. of Strain (%) | Sub-total (%) | No. of Strain (%) | Sub-total (%) | No. of Strain (%) | Sub-total (%) |
| **Against two** | [1]AMP-10, [2]CH-30 | 2(8.3) | 2(8.3) | - | - | - | - |
| | CH-30, [3]TE-30 | - | - | 2(22.2) | 2(22.2) | - | |
| **Against three** | [4]CTX-30, CH-30, [5]C-30 | 1 (4.2) | 7(29.2) | - | 3(33.3) | - | - |
| | CTX-30, CH-30, TE-30 | 2(8.3) | | - | | - | |
| | CTX-30, CH-30, [6]DA-10 | 1 (4.2) | | - | | - | |
| | AMP-10, CH-30, [7]CIP-10 | 1 (4.2) | | - | | - | |
| | AMP-10, CH-30, DA-10 | 1 (4.2) | | - | | - | |
| | AMP-10, CH-30, C-30 | - | | 2(22.2) | | - | |
| | AMP-10, /[8]CFM-5, CTX-30/, CH-30 | 1 (4.2) | | - | | - | |
| | AMP-10, /CFM-5, CTX-30, [9]CRO-30/, CH-30 | - | | 1(11.1) | | - | |
| **Against four** | CTX-30 CH-30, [10]SXT-25, DA-10 | 1 (4.2) | 5(20.8) | - | 4(44.4) | - | - |
| | AMP-10, CTX-30, CH-30, DA-10 | 1 (4.2) | | - | | - | |
| | AMP-10, CH-30, C-30, SXT-25 | 1 (4.2) | | - | | - | |
| | AMP-10, CTX-30, CH-30, C-30 | 1 (4.2) | | - | | - | |
| | AMP-10, CH-30, C-30, DA-10 | - | | 1(11.1) | | - | |
| | CFM-5, CTX-30, CH-30, TE-30 | - | | 1(11.1) | | - | |
| | AMP-10, CTX-30, CH-30, TE-30 | - | | 1(11.1) | | - | |
| | /CTX-30, CRO-30/, CH-30, C-30, TE-30 | 1 (4.2) | | - | | - | |
| | AMP-10, /CFM-5, CTX-30/, CH-30, C-30 | - | - | 1(11.1) | | - | |
| **Against five** | CFM-5, CH-30, C-30, TE-30, DA-10 | 1 (4.2) | 7(29.2) | - | - | - | 1(20.0) |
| | AMP-10, CTX-30, CH-30, C-30, DA-10 | 1 (4.2) | | - | | - | |
| | CTX-30, CH-30, C-30, CIP-10, DA-10 | 1 (4.2) | | - | | - | |
| | CH-30, C-30, SXT-25, TE-30, DA-10 | 1 (4.2) | | - | | - | |
| | CFM-5, CH-30, C-30, SXT-25, TE-30 | 1 (4.2) | | - | | - | |
| | AMP-10, CTX-30, CH-30, TE-30, NA-30 | - | | - | | 1(20.0) | |
| | AMP-10, /CFM-5, CTX-30/, CH-30, CIP-10, DA-10 | 1 (4.2) | | - | | - | |
| | AMP-10, /CFM-5, CTX-30/, CH-30, C-30, TE-30 | 1 (4.2) | | - | | - | |
| **Against six** | AMP-10, CFM-5, CH-30, SXT-25, TE-30, DA-10 | 1 (4.2) | 1(4.2) | - | - | - | 4(80.0) |
| | AMP-10, CTX-30, CH-30, C-30, TE-30, [11]NA-30 | - | | - | | 1(20.0) | |
| | AMP-10, CFM-5, CH-30, C-30, TE-30, NA-30 | - | | - | | 1(20.0) | |
| | AMP-10, /CFM-5, CTX-30, CRO-30/, CH-30, TE-30, NA-30 | - | | - | | 1(20.0) | |
| | AMP-10, /CFM-5, CTX-30/, CH-30, TE-30, DA-10, NA-30 | - | | - | | 1(20.0) | |

[1]Ampicillin (AMP-10),

[2]Cephalothin (CH-30),

[3]Tetracycline (TE- 30),

[4]Cefotaxime (CTX-30),

[5]Chloramphenicol (C-30),

[6]Clindamycin (DA 10),

[7]Ciprofloxacin (CIP 10),

[8]Cefixime (CFM-5),

[9]Ceftriaxone (CRO-30),

[10]Sulphamethazole Trimethoprim (SXT 25),

[11]Nalidixic Acid (NA 30)

The second highest prevalence of thermophilic *Campylobacter* species was isolated from carcass/meat swab in this study. Study conducted in Mekele, Northern Ethiopia also reported higher prevalence (11.9%) of *Campylobacter* species isolates from cattle meat [17]. This could be due to *Campylobacter* species are the normal flora of gastrointestinal tract [32, 33] and in case of carcass swabs, it could be due to contamination of carcass with intestinal contents during manual skinning, evisceration, carcass washing and processing at abattoir. Similarly, *Campylobacter* species were isolated from hand and knife swabs, and it could be due to cross contamination due to negligence of hand and knife washing in between cattle slaughtering. Personnel hands and utensils such as knife and cutting boards are among significant contributors for the contamination of animal food products with *Campylobacter* species [3–5]. Even though no statistically significant difference was observed, the number of *Campylobacter* species isolated from knives swabs was higher than personnel hands swabs. Study conducted in Denmark also reported possible *Campylobacter* species cross contamination of meat products from workers' hands and stabbing knives [34]. Therefore, abattoir management should focus on hygiene of slaughtering houses by avoiding free movement, washing hands and knife after slaughtering each cattle and improving water supply. As well as, implementing regular inspection in abattoirs for infectious diseases including *Campylobacter* species is mandatory at national level. Moreover, awareness creation or training should be given for abattoir workers to minimize or prevent cross contamination and the risk of zoonotic food borne infectious diseases transmission to human.

The predominant thermophilic *Campylobacter* species isolate in the current study was *C. jejuni* followed by *C. coli* and *C. lari*, respectively. Previous study conducted in Jimma Zone documented similar findings but the number of *C. lari* is higher in the current study, 13.2% versus 3.1% [15]. This might be due to increased drug resistance pattern of *C. lari* observed in this study.

An increasing numbers of *Campylobacter* isolates have developed resistance to antimicrobials, mainly in low and middle-income countries. In recent years, increased resistance was observed for fluoroquinolones, macrolides, aminoglycosides, and beta lactams [35]. Furthermore, intrinsic resistance in *C. jejuni* and *C. coli* has been described against penicillins, most of the cephalosporin as well as trimethoprim, sulfamethoxazole, rifampicin, and vancomycin [36–39]. *Campylobacter* isolates in the present study were susceptible to Nalidixic acid (86.8%), ciprofloxacin (86.8%), Sulphamethazole (84.2%) and ceftriaxone (78.9%). However, *Campylobacter* species in this study were highly resistant for chloramphenicol and Tetracycline. Studies conducted in Morocco also documented resistant strains of Campylobacter species for tetracycline and other regimen of antibiotics [39, 40]. Other studies also reported chloramphenicol and Tetracycline resistant *Campylobacter* species [16, 41]. Tetracycline has been used for animal production for many years, and this study also showed high tetracycline resistance in all species of *Campylobacter* isolates. The resistance to tetracycline in *Campylobacter* species might be due to presence of plasmid-encoded tet (O) resistant gene in both *C. jejuni* and *C. coli* [38, 39]. *Campylobacter* species isolates in this study were also highly resistant to Cephalothin, Ampicillin and cefotaxime. Other studies conducted elsewhere also documented resistance of *Campylobacter* isolates for these drugs [16, 17, 42]. This might be alarm for the future serious challenges in the treatment of human campylobacteriosis associated with food of animal origins.

Multidrug resistance was observed in all *Campylobacter* isolates in the present study. all isolates of *C. lari* showed particularly, the highest multidrug resistance pattern, against five to six drug classes, followed by *C. jejuni*, against three to five drug classes. The finding was in agreement with the result documented in Gambella, Ethiopia [16]. Several investigators have also reported the increasing incidence of high multidrug resistance of *Campylobacter* species [39,

43, 44]. Multiple drug resistance of thermophilic *Campylobacter* species might be increased due to transmission of the bacteria from human to animal and vice versa, and these expose the bacterium to different antimicrobial treatments utilized for both animal and human.

The limitation of this study was lack of molecular techniques for identification of Campylobacter species and for the detection of responsible genes for the developed drug resistance. Moreover, we did not used standard strains of Campylobacter species; instead we used *E. coli* (ATCC 25922) to check the contents of drugs.

## Conclusion

Three species of *Campylobacter* identified were mainly isolated from faecal samples. Most of the isolates were resistant to Cephalothin, Ampicillin and Cefotaxime. Multidrug resistance among the isolates was also high. Detection of *Campylobacter* species from the faeces, carcass, knife and personnel shows possible cross-contamination during the slaughtering process. This may increase the risks of *Campylobacter infection* to people consuming uncooked meat. High proportion of multidrug resistance among the isolates may also result in horizontal spread of drug resistance in human and may pose a threat to humans and further limits therapeutic options. Based on the findings; it is sensible to recommend regular coordinated actions like judicial antibiotic usage and hazard analysis of critical control points from the farm, through the abattoir to the retailer and awareness creation on good hygienic practice and processing of raw meat to minimize or eliminate the risk of infectious agents in general and *Campylobacter* contamination in particular.

## Supporting information

**S1 File.**
(DOCX)

## Acknowledgments

We thank Jimma Town Abattoir workers for their support and participation in this study. We are also thankful to technical staff of School of Veterinary Medicine for their assistance during laboratory works and Jimma University College of Agriculture and Veterinary Medicine.

## Author Contributions

**Conceptualization:** Motuma Debelo.

**Data curation:** Motuma Debelo, Nezif Mohammed, Abebaw Tiruneh.

**Formal analysis:** Motuma Debelo, Nezif Mohammed, Abebaw Tiruneh, Tadele Tolosa.

**Funding acquisition:** Motuma Debelo, Nezif Mohammed, Abebaw Tiruneh, Tadele Tolosa.

**Investigation:** Motuma Debelo, Nezif Mohammed, Tadele Tolosa.

**Methodology:** Motuma Debelo, Nezif Mohammed, Abebaw Tiruneh, Tadele Tolosa.

**Project administration:** Motuma Debelo, Tadele Tolosa.

**Resources:** Motuma Debelo, Nezif Mohammed, Abebaw Tiruneh, Tadele Tolosa.

**Software:** Motuma Debelo, Nezif Mohammed, Abebaw Tiruneh.

**Supervision:** Motuma Debelo, Tadele Tolosa.

**Validation:** Motuma Debelo, Nezif Mohammed, Abebaw Tiruneh, Tadele Tolosa.

**Visualization:** Motuma Debelo, Nezif Mohammed, Abebaw Tiruneh, Tadele Tolosa.

**Writing – original draft:** Motuma Debelo, Nezif Mohammed.

**Writing – review & editing:** Motuma Debelo, Nezif Mohammed, Abebaw Tiruneh, Tadele Tolosa.

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
