## [Decision Letter · Decision Letter 0]

8 Aug 2022

PONE-D-22-17337Isolation, identification and antibiogram profile of thermophilic Campylobacter species from bovine, knives and personnel at Jimma Town abattoir, EthiopiaPLOS ONE

Dear Dr. Debelo,

Thank you for submitting your manuscript to PLOS ONE. After careful consideration, we feel that it has merit but does not fully meet PLOS ONE’s publication criteria as it currently stands. Therefore, we invite you to submit a revised version of the manuscript that addresses the points raised during the review process.

 Please submit your revised manuscript by Sep 22 2022 11:59PM. If you will need more time than this to complete your revisions, please reply to this message or contact the journal office at plosone@plos.org. Please include the following items when submitting your revised manuscript:A rebuttal letter that responds to each point raised by the academic editor and reviewer(s). You should upload this letter as a separate file labeled 'Response to Reviewers'.A marked-up copy of your manuscript that highlights changes made to the original version. You should upload this as a separate file labeled 'Revised Manuscript with Track Changes'.An unmarked version of your revised paper without tracked changes. You should upload this as a separate file labeled 'Manuscript'.

We look forward to receiving your revised manuscript.

Kind regards,

Monica Cartelle Gestal, PhD

Academic Editor

PLOS ONE

Journal Requirements:

 "Authors who received each award; Motuma Debelo, Nezif Mohammed,

Abebaw Tiruneh and Tadele Tolosa

No Specific Grant numbers awarded to each author.

This research work was funded by Jimma University College of Agriculture and Veterinary Medicine.

The funders had no role in study design, data collection and analysis, decision to

publish, or preparation of the manuscript."

3. "Thank you for stating the following in the Acknowledgments Section of your manuscript: 

   "We thank Jimma Town abattoir workers for their support and participation in this study. We are also thankful to technical staff of School of Veterinary Medicine for their assistance during laboratory works and Jimma University, College of Agriculture and Veterinary Medicine for their facilitation and some funds."

 "Authors who received each award; Motuma Debelo, Nezif Mohammed,

Abebaw Tiruneh and Tadele Tolosa

No Specific Grant numbers awarded to each author.

This research work was funded by Jimma University College of Agriculture and Veterinary Medicine.

The funders had no role in study design, data collection and analysis, decision to

publish, or preparation of the manuscript."

5. Please ensure that you refer to Figure 1 in your text as, if accepted, production will need this reference to link the reader to the figure.

6. Please include a caption for figure 1. 

Additional Editor Comments:

As you can see on the reviewers comments, it was some controversy with this manuscript. Please pay special attention to the comments provided by reviewer 1 and 3 prior to resubmission. 

Thanks

Reviewers' comments:

Reviewer's Responses to Questions

**Comments to the Author**

1. Is the manuscript technically sound, and do the data support the conclusions?

Reviewer #1: No

Reviewer #2: Yes

Reviewer #3: Partly

2. Has the statistical analysis been performed appropriately and rigorously? 

Reviewer #1: N/A

Reviewer #2: Yes

Reviewer #3: Yes

3. Have the authors made all data underlying the findings in their manuscript fully available?

Reviewer #1: Yes

Reviewer #2: Yes

Reviewer #3: Yes

4. Is the manuscript presented in an intelligible fashion and written in standard English?

Reviewer #1: No

Reviewer #2: Yes

Reviewer #3: Yes

5. Review Comments to the Author

Reviewer #1: Comments to the Author:

The paper described the “Isolation, identification and antibiogram profile of thermophilic Campylobacter species from bovine, knives and personnel at Jimma Town abattoir, Ethiopia”.

Major comment: After carefully reading this paper, it was observed that the findings of the paper are of local interest and importance. Other readers may not be benefited by the data. Further, the study carried out at only one abattoir and therefore the limited number of isolates (n=38) obtained in this study are not sufficiently addressed to draw any logistic conclusion. The article does not have any novelty of data or any molecular characterization of isolates. Further, the data of the study is also not presented well in the manuscript.

Keeping in view of the facts, the article may not be found suitable for publication in the reputed and high impact Plos One journal.

XXXXXXXX

Minor comments :

The manuscript presents the research to understand the current status of Antimicrobial Resistance (AMR) in Campylobacter isolates from faecal samples of bovine, carcases of slaughtered animals, knives and personnel employed at abattoir.

Overall, the manuscript requires editing on the grammar at many places to improve the clarity and data presentation. Some sentences are not clear and difficult to understand. Objective to rephrase, not clear, and to show novelity or improvement from previous studies carried in this era.

In Methodology, what i am missing is the absence of molecular study for identification of different species and for detection of different AMR and Virulence genes.

Emphasize the new and important aspects of your study and put your findings in the context of the totality of the relevant evidence.

I didn’t find any standard strains of campylobacter( be it jejuni, coli, lari) were used for this study?

As a general comment, no limitations of the study were given in manuscript. In my mind the major limitation is the absence of genotypic studies for AMR and virulence of isolates it could have made your study more meaningful and interesting.

The objective of this manuscript was to estimate the occurrence and antimicrobial resistance of thermophilic Campylobacter strains isolated from from faecal samples of bovine, carcases of slaughtered animals, knives and personnel employed at abattoir in Ethiopia. There are many articles with similar objectives and I do not observe in this any original element beyond the region studied. However, this fact seems to me not to be enough and perhaps the authors should consider a journal with local impact.

Reviewer #2: Review Report:

Line 15 – make the campylobacter italics “Campylobacter species from bovine” make.

Line 18 – make the italics “in vitro”

Line 23 – Check the spelling “specious”

Line 76 --- Looking at the sample data, the samples have not been collected on a regular basis to estimate the prevalence, and it does not comply with cross-sectional study norms! Again please correct for grammatical errors.

The manuscript has been looked at for minor grammar mistakes and the manuscript suffers from a huge amount of grammatical errors. This manuscript has to be handed over a native English speaker and revised accordingly.

Reviewer #3: The authors should improved their manuscript following the reviewer comment uploaded. The authors should improved their manuscript following the reviewer comment uploaded.The authors should improved their manuscript following the reviewer comment uploaded.The authors should improved their manuscript following the reviewer comment uploaded.

6. PLOS authors have the option to publish the peer review history of their article (what does this mean?). If published, this will include your full peer review and any attached files.

Reviewer #1: **Yes: **Bilal Ahmad Malla

Reviewer #2: No

Reviewer #3: No

---

## [Author Response · Author response to Decision Letter 0]

20 Sep 2022

Reviewer #1

1. We thank the reviewer for the fruitful comments

Major comment: 

a. We the authors agree with the reviewer comment that it was better if the study was done at national level. However, Jimma Town is not as such small town and it is home for more than 207,573 (projected population in 2021/not actual) with 50.52km2 in south-western Ethiopia. There is only one abattoir in the town delivering service for the vast population in the town. The research finding will benefit the local and national government to take measures in Jimma abattoir in particular and abattoirs in the country in general. Researchers/scientific community will be benefitted to prepare review articles, taking this findings as a baseline research and to consider the aggravation of antimicrobial resistance pattern both in human and animal populations. To make it very clear to the international readers, we elaborated the background of our study setting.

b. It was better if the number of isolates obtained (n=38) was increased. However, our finding is consistent (relatively high) with similar previous researches done in Ethiopia.

c. We strongly agree with the comment not to use molecular characterization as diagnostic method and we included in the study limitation. It was also our plan and tried our best to get primers from abroad, but we missed due to COVID-19 pandemic lockdown in the world.

Minor comment

a. We thank the reviewer comment to edit the grammar and sentence structure. We improved the grammar and unclear sentences in the revised manuscript.

b. We thank the reviewer for the comment ‘absence of molecular study’. We took as limitation of the study.

c. We thank the reviewer comments on the objectives. We improved/rephrased our objectives in the revised manuscript.

d. We thank the reviewer for the comment. We included the importance of our finding in the revised manuscript.

e. We thank the reviewer for the comment to use standard strain of Campylobacter. We could not find standard strain for Campylobacter species from the national laboratory and we have used E. coli (ATCC 25922) as standard and included in the limitation of the study in the revised manuscript.

f. We thank the reviewer for the comment and the revised manuscript included ‘limitations of the study’.

g. We respect the reviewer comment and strongly disagree with the idea ‘study region is not enough reason to conduct research’. There are millions of prevalence studies for the same disease in different regions and/or time frame. In previous works, prevalence as well AMR was only determined from faecal but in our case it was from faecal, carcasses, personnel hands and knife samples. Therefore, our study is not out of this fact and we believe it is very important and respected to show the gaps and identify AMR Campylobacter strains in the region. 

Reviewer #2

a. We thank the reviewer for the comments Line 15,18 and 23. We corrected accordingly in the revised manuscript.

b. We thank the reviewer comment on study design. We trimmed the full description of sample collection methods to make it shorter previously. It is included now (the regular basis) in the revised manuscript.

c. We thank the reviewer comment for the English language. Senior professors edited the English language of our revised manuscript.

Reviewer #3

a. We thank the reviewer comment on the tittle of the manuscript. We reformulated the tittle as ‘Isolation, identification and antibiotic resistance profile of thermophilic Campylobacter species from bovine, knives and personnel at Jimma Town abattoir, Ethiopia’.

b. We thank the reviewer for the comment on the English language improvement and our revised manuscript is edited by senior professors.

c. We thank the reviewer comment on capitalizing genus name initials in the manuscript and corrected accordingly.

d. We thank the reviewer for the comment to include molecular techniques (PCR) for confirmation of Campylobacter species. We also strongly agree with the comment, and there was no Campylobacter primer in Ethiopia and difficult to get from abroad due to COVID-19 pandemic at a time. We included in the limitation of the study in the revised manuscript.

e. We appreciate and thank the reviewer for suggesting very important research findings to compare with our work. We discussed our findings comparing with suggested studies.

f. We thank the reviewer comment and the discussion part was improved in the revised manuscript.

g. We thank the reviewer for critical review and the mentioned words in line 35 & 36 were eliminated in the revised manuscript.

h. We thank the reviewer for the comment and some points were added after reference (1,2) in the revised manuscript.

i. We thank the reviewer for the comments and paragraphs in line 38-39, 81-83 and 88-89 were reformulated in the revised manuscript.

j. We thank the reviewer for the comments and words mentioned in line 93 and 96 were replaced accordingly.

Authors’ response to academic editor

We thank the academic editor for the detailed comments on the format of the manuscript and other comments.

1. The manuscript was checked according to the journal formats.

2. The funding section was revised based on the comments given in the revised manuscript.

3. The acknowledgement part is corrected in the revised manuscript.

4. Supporting files were uploaded as indicated by the academic editor.

5. The authors prefer to display tables than figure, and figure 1 is deleted from the revised manuscript.

---

## [Decision Letter · Decision Letter 1]

11 Oct 2022

Isolation, Identification and Antibiotic Resistance Profile of Thermophilic Campylobacter Species from Bovine, Knives and Personnel at Jimma Town Abattoir, Ethiopia

PONE-D-22-17337R1

Dear Dr. Debelo,

We’re pleased to inform you that your manuscript has been judged scientifically suitable for publication and will be formally accepted for publication once it meets all outstanding technical requirements.

Kind regards,

Monica Cartelle Gestal, PhD

Academic Editor

PLOS ONE

Additional Editor Comments (optional):

Reviewers' comments:

Reviewer's Responses to Questions

**Comments to the Author**

1. If the authors have adequately addressed your comments raised in a previous round of review and you feel that this manuscript is now acceptable for publication, you may indicate that here to bypass the “Comments to the Author” section, enter your conflict of interest statement in the “Confidential to Editor” section, and submit your "Accept" recommendation.

Reviewer #2: All comments have been addressed

Reviewer #3: All comments have been addressed

2. Is the manuscript technically sound, and do the data support the conclusions?

Reviewer #2: Yes

Reviewer #3: Yes

3. Has the statistical analysis been performed appropriately and rigorously? 

Reviewer #2: Yes

Reviewer #3: Yes

4. Have the authors made all data underlying the findings in their manuscript fully available?

Reviewer #2: Yes

Reviewer #3: Yes

5. Is the manuscript presented in an intelligible fashion and written in standard English?

Reviewer #2: Yes

Reviewer #3: Yes

6. Review Comments to the Author

Reviewer #2: The manuscript has been revised and as per the suggestion of the reviewer. So it may be accepted for publication.

Reviewer #3: accepte accepte accepte accepte accepte accepte accepte accepte accepte accepte accepte accepte accepte accepte accepte accepte accepte accepte accepte accepte accepte accepte accepte accepte accepte accepte accepte accepte accepte accepte accepte accepte accepte accepte accepte accepte accepte accepte accepte accepte accepte accepte accepte accepte accepte accepte accepte accepte accepte accepte accepte accepte accepte accepte

7. PLOS authors have the option to publish the peer review history of their article (what does this mean?). If published, this will include your full peer review and any attached files.

Reviewer #2: **Yes: **Amit Kumar Verma

Reviewer #3: No

---

## [Editor Report · Acceptance letter]

14 Oct 2022

PONE-D-22-17337R1 

Isolation, Identification and Antibiotic Resistance Profile of Thermophilic *Campylobacter* Species from Bovine, Knives and Personnel at Jimma Town Abattoir, Ethiopia  

Dear Dr. Debelo:

I'm pleased to inform you that your manuscript has been deemed suitable for publication in PLOS ONE. Congratulations! Your manuscript is now with our production department. 

Kind regards, 

on behalf of

Dr. Monica Cartelle Gestal 

Academic Editor

PLOS ONE